# Wild Terrestrial Animal Re-Identification Based on an Improved Locally Aware Transformer with a Cross-Attention Mechanism

**DOI:** 10.3390/ani12243503

**Published:** 2022-12-12

**Authors:** Zhaoxiang Zheng, Yaqin Zhao, Ao Li, Qiuping Yu

**Affiliations:** College of Mechanical and Electronic Engineering, Nanjing Forestry University, Nanjing 210037, China

**Keywords:** wild terrestrial animal re-identification, cross-attention mechanism, locally aware transformer, local feature differences

## Abstract

**Simple Summary:**

The re-identification of animals can distinguish different individuals and is regarded as the premise of modern animal protection and management. The re-identification of wild animals can be inferred and judged by the difference in their coat colors and facial features. Due to the limitation of long-distance feature extraction, CNN (Convolutional Neural Network) is not conducive to mining the relationships among local features. Therefore, this paper proposes a transformer network structure with a cross-attention block (CAB) and local awareness (CATLA transformer) for the re-identification of wild animals. We replace the self-attention module of the LA transformer with CAB to better capture the global information of the animal body and the differences in facial features, or local fur colors and textures. According to the distribution of animal body parts of the animal standing posture, we redesigned the layer structure of the local aware network to fuse the local and global features.

**Abstract:**

The wildlife re-identification recognition methods based on the camera trap were used to identify different individuals of the same species using the fur, stripes, facial features and other features of the animal body surfaces in the images, which is an important way to count the individual number of a species. Re-identification of wild animals can provide solid technical support for the in-depth study of the number of individuals and living conditions of rare wild animals, as well as provide accurate and timely data support for population ecology and conservation biology research. However, due to the difficulty of recording the shy wild animals and distinguishing the similar fur of different individuals, only a few papers have focused on the re-identification recognition of wild animals. In order to fill this gap, we improved the locally aware transformer (LA transformer) network structure for the re-identification recognition of wild terrestrial animals. First of all, at the stage of feature extraction, we replaced the self-attention module of the LA transformer with a cross-attention block (CAB) in order to calculate the inner-patch attention and cross-patch attention, so that we could efficiently capture the global information of the animal body’s surface and local feature differences of fur, colors, textures, or faces. Then, the locally aware network of the LA transformer was used to fuse the local and global features. Finally, the classification layer of the network realized wildlife individual recognition. In order to evaluate the performance of the model, we tested it on a dataset of Amur tiger torsos and the face datasets of six different species, including lions, golden monkeys, meerkats, red pandas, tigers, and chimpanzees. The experimental results showed that our wildlife re-identification model has good generalization ability and is superior to the existing methods in mAP (mean average precision), and obtained comparable results in the metrics Rank 1 and Rank 5.

## 1. Introduction

With human society’s industrial and agricultural development, wild animals and plants have gradually lost their living spaces. *The Convention on Biological Diversity* [1] was proposed to protect the fragile field environment. However, according to the “Red List” of threatened species of the International Union for the Conversation of Nature (IUCN) in 2008 [2], global biodiversity is still seriously threatened.

In order to change the living status of animal species, we need to know their populations, distributions and behavior. The re-identification of animals can distinguish different individuals and is regarded as the premise of modern animal protection and management. Meanwhile, for animals in the zoo, re-identification can help staff to establish their archives and analyze their growth and breeding behaviors, in order to reasonably plan their daily lives. For animals in the wild, individual identification can assist researchers in knowing their health status, and studying their lifestyle and distribution, which provides a factual basis for making appropriate protection measures. Re-identification methods have been used to assess animal population size and density. For example, Shannon Gowans [3] recorded whale photos for 11 years in order to study the population size of northern bottlenose whales in Canadian waters. According to the permanent characteristics, transient characteristics, and behavior of black bears, Melissa Reynolds-Hogland [4] analyzed the density, age, and sex of black bears in western Montana. Peter Kulits [5] built a platform and database based on a deep learning network, which combined manual tagging and computer vision algorithms, helping to monitor the population of elephants in the Greater Maasai Mara ecosystem.

Over the years, camera traps [6] have been used to take a large number of pictures without disturbing animals, making image-based animal research possible. The study targets in this paper were all large warm-blooded animals, so we chose common passive infrared trigger cameras to record the animal images [7]. A passive infrared trigger camera can sense the change of heat through its infrared sensor when a warm-blooded animal is passing, so the animal will be automatically photographed after the camera is triggered. Generally, the infrared trigger camera is fixed according to the shape of the target animals, and it is usually bundled at 0.5~1.3 m of the tree trunk. A flat and open terrain is selected as the camera site, and there are few shrubs and grasses at the range of 3 m in front of the lens, within the camera angle of 30°. Furthermore, the photos taken by the infrared trigger camera can also be transmitted to researchers through wireless transmission.

In the early years, Meek et al. [8] once invited some mammalians to identify individual animals from photos. This method not only depends on the professional level of the discriminator but also has high error rates and a time-consuming nature. To avoid the above problems, researchers applied machine learning methods for the identification of individual wild animals. For example, XU et al. [9] and Arzoumanian et al. [10] identified individuals based on their body surfaces, such as stripes, spots, and scar features. However, the methods are designed according to the body surface features of a certain species, and thus cannot be generalized to other species.

With the wide application of deep learning technology in the field of image processing [11,12,13], researchers have tried to use deep learning networks for the re-identification of wild animals. Carter et al. [14] developed the first tool for individual re-identification of animals, MYDAS, which searched the most similar individuals from a database of green sea turtles. Nepovinnykh et al. [15] used the Siamese neural network to recognize individual seals. Yu et al. [16] and Li et al. [17] applied the convolutional networks ResNet and DenseNet to realize the re-identification of Amur tigers. The re-identification of primates [18,19,20,21] has also achieved some progress. The methods extracted face features of primates based on convolutional neural network (CNN) of human face recognition. For example, Debayan et al. [18] used group convolution to improve the traditional CNN structure to reduce the number of parameters. Guo et al. [19] developed the model Tri-AI, which can quickly detect and recognize the faces of golden monkeys based on Fast RCNN. In addition, Schofield, D [20] and Freytag, A [21] both use CNN to recognize the chimpanzees by extracting their facial features.

Due to the limitations of long-distance feature extraction, a CNN is not conducive to mining the relationships among local features. In addition, due to the shy characteristics of wild animals and the complex field environment, it is difficult to capture whole bodies of animals from multiple angles, which also poses a challenge to the re-identification of wild animals. Transformer, as an emerging deep learning structure, can utilize multi-head self-attention to capture long-distance relationships and pay attention to local features, so it has already shone in the field of human re-identification [22,23,24]. However, the standing posture of humans is quite different from that of wild animals, so the methods of human re-identification cannot be directly applied to wild animal re-identification. Therefore, this paper proposed a transformer network structure with a cross-attention block (CAB) and local awareness (CATLA transformer) for the re-identification of wild animals. We replaced the self-attention module of the LA-Transformer with CAB to better capture the global information of the animal body and the differences in facial features, or local fur colors and textures. According to the distribution of animal body parts of the animal standing posture, we redesigned the layer structure of the local aware network to fuse the local and global features. The specific contributions are as follows:In order to better extract and fuse global and local information on wildlife, we first applied the transformer network structure to the re-identification of wildlife and proposed a transformer network based on a cross-attention mechanism for the re-identification of wild terrestrial animals;After partitioning the whole image into patches, in order to extract the local features of the patches and the global correlation between patches, we replaced the self-attention module of the LA transformer with CAB, which captures the global information of the animal appearance and the differences in local features, such as local fur colors and textures;At the stage of the feature fusion, the hierarchical structure of the locally aware network was redesigned according to the distribution of animal body parts in the standing posture. After fusing the weighted-average values of global and local tokens, we obtained the globally enhanced local tokens where the fused features were arranged into 7 × 28 2D distribution;To validate the generalization ability of the model, we tested different types of datasets, such as the animal trunk dataset and the animal face datasets including those of tigers, lions, golden monkeys, and other common species.

## 2. Materials and Methods

In this section, we mainly introduce the datasets in the experiments, and the division of the training set and the testing set, then we give the pipeline of the wildlife recognition network.

### 2.1. Datasets

In order to verify the performance of our proposed model from different perspectives, we chose three public datasets that are widely used for animal re-identification. These datasets include animal trunk and face images, RGB, and gray images.

The dataset ATRW [17] was first proposed to study the re-identification of Amur tigers and monitor their number and distribution in the wild. To the best of our knowledge, ATRW is the only public dataset with whole animal body images. As the testing set is not released, for our experiments we chose the images in the training set. The dataset Tri-AI [19] has individual animals’ faces for six species. The pictures of meerkats, lions, and red pandas were captured during the daytime, whereas those of golden monkeys and tigers were captured at night. We also used two challenging datasets [21], C-Zoo and C-Tai, that have chimpanzees’ faces under complex interference environments. The two datasets included 102 chimpanzee IDs (individuals). We deleted 10 IDs with a few images and retained the remaining 92 IDs. The datasets have the facial images of the same chimpanzee individuals at different ages from different points of view, and some different chimpanzee individuals in the datasets almost have the same facial features, which is challenging for the re-identification task. We also expanded the dataset by flipping the images.

The details of the datasets are shown in Table 1, and exemplary images from each dataset are shown in Figure 1.

### 2.2. Methods

As shown in Figure 2, our study was divided into two processes, namely, training the network and testing. In the training stage, a series of animal images were input into the CATLA transformer encoder module to extract local features and global information. It can be seen in Figure 2 that CATLA paid more attention to the stripes on the tiger’s back. Then, the local awareness network was used to fuse the local features and global information. Finally, the classification layer was applied for individual recognition. In the testing process, the Query images (unknown IDs) and the Gallery images (known IDs) were input into the trained network. After feature extraction and classification, the specific identity of each tiger in Query was obtained.

Figure 3 shows the overall pipeline of our approach. We constructed a transformer network with cross-attention and local awareness (CATLA transformer) for the re-identification of wild terrestrial animals. Our model consists of three parts, namely, the embedding layer [25], the CATLA transformer encoder module and the local awareness network [26]. The embedding layer is used to flatten an H×W×C 2D image into an $N\times \left(P^{2}\times C\right)$ sequence, where *P* is the size of the image patch; N is the number of image blocks, which is added with the position information and input into the transformer encoder module. The cross-attention block in the transformer encoder module combines the inner-patch attention with the cross-patch attention. It cannot only capture the local differences in animal fur or facial features, but also obtain global information on animal body appearance. The multilayer perceptron block [27] fuses the global token into local tokens using the weighted-average method, which effectively enhances and fuses multi-scale spatial features. Finally, the features of each layer are input into the FC classifier and softmax module of the local perceptual network to identify the individual animal.

#### 2.2.1. CATLA Transformer Encoder for Extracting Wildlife Muti-Scale Features

Cross-attention blocks (CAB) [28] are the main blocks of the CATLA transformer encoder. The CAB structure uses inner-patch self-attention (IPSA) and cross-patch self-attention (CPSA) to realize the attention calculation among image patches and feature maps of each channel, respectively, and fuse the inner-patch (local features) and cross-patch (global information) attention.

As shown in Figure 4a, the IPSA module unfolds all channel inputs in each patch instead of the whole image and reshapes these patches of all channels into the original shape after computing the IPSA attention of pixels. Therefore, the IPSA module was used to extract local features of wild animal fur or faces. However, IPSA can only obtain the relationship between the pixels of a patch and cannot obtain the associated information between patches. We used the CPSA module, as shown in Figure 4b, to unfold the input of a single channel to calculate the correlation characteristics between patches and obtain the global information of the whole image, and then reshaped these patches of the channel into the original shape.

IPSA and CPSA build cross-attention by stacking basic modules. The stacking mode is shown in Figure 5. The stacking IPSA and CPSA are built with layer normalization (LN), multilayer perceptron (MLP), and shortcut connection. The whole process is shown by Formulas (1)–(6).
(1)$$f_{1}=feature+IPSA\left(LN\left(feature\right)\right)$$
(2)$$f_{1}^{\prime }=f_{1}+MLP\left(LN\left(f_{1}\right)\right)$$
(3)$$f_{2}=f_{1}^{\prime }+CPSA\left(LN\left(f_{1}^{\prime }\right)\right)$$
(4)$$f_{2}^{\prime }=f_{2}+MLP\left(LN\left(f_{2}\right)\right)$$
(5)$$f_{3}=f_{2}^{\prime }+IPSA\left(LN\left(f_{2}^{\prime }\right)\right)$$
(6)$$Feature=f_{3}+MLP\left(LN\left(f_{3}\right)\right)$$
where feature denotes that the input features of the channel, and $f_{i}^{’\;}$ is an output of one block MLP with LN. $f_{j}$ is the output of one block (CSPA or ISPA) with LN. Feature presents the output of the cross-attention block.

#### 2.2.2. Locally Aware Network

We calculated the weighted average of local features and global features based on the locally aware network [29] and then performed the average pooling operation to obtain the features of each layer. The weighted average formula is as follows: let the characteristics of each layer pass through FC layer, softmax layer and argmax layer to obtain the prediction results.
(7)$$L_{i}=\frac{1}{N_{R}}\overset{\left(i+1\right)\times N_{R}}{\underset{j=i\times N_{R}+1}{\sum }}\frac{\left(Q_{j}+\lambda G\right)}{\left(1+\lambda \right)}$$
(8)$$\mathrm{score}=\sum_{i=0}^{N_{R}}softmax\left(FC\left(L_{i}\right)\right)$$

(9)*Prediction = argmax(score)*$i=0,\cdots N_{C}-1$, where $N_{R}$ and $N_{C}$ are the number of layers and the number of patches in each column; $N_{R}\;$= $\frac{\;1}{4}N_{C}\;$=$\;\frac{1}{2}\sqrt{N}$, where N is the total number of patches. The function FC(·) represents fully connected layers, and softmax(·) is an activation Function. After the softmax scores are summed together, argmax(·) obtains the argument of the maximum score that represents the ID of the animal [29].

Compared with the human body, the movement of animal body parts is relatively simple. Therefore, we redesigned the layer structure of the locally aware network to reduce the intra-layer difference and broaden the inter-layer difference. After the weighted-average fusion of global and local tokens, the fused feature elements were arranged into 7 × 28 2D distributions, and the number of layers was reduced.

## 3. Results

### 3.1. Experimental Details

The program used in this study was written in Pytorch and trained on a computer configured with an Intel i7-11800H CPU and NVIDIA GeForce RTX 3060 GPU. The numbers of images in the training subset and the testing subset are shown in Table 1. During training the backbone on the image dataset, the initial learning rate was set to 0.0003, and the learning rate was multiplied by 0.9 to reduce it every two epochs. We trained every network for 30 epochs. The size of all images input into the network was fixed to 224 × 224, The size of the patch was 16 × 16, which was divided into 196 small patches in total, and the number of channels was three. Therefore, the dimensions of each small patch were 16 × 16 × 3 = 768. The number of layers of the locally aware network were set as seven; therefore, the features of 196 patches were arranged into 7 × 28 2D distributions. The backbone network of the CATLA transformer is the vision transformer (VIT), but VIT requires a large number of images to be trained effectively, so we directly adopted the pre-training model of VIT on ImageNet and used our dataset to fine-tune the pre-training model.

Re-identification is generally evaluated by two metrics, cumulative matching features (CMC) and mean average precision (mAP). When calculating the evaluation index, it is necessary to divide the test set into the Query image set and the Gallery image set. The re-ID methods select an image in the Query image set and retrieve the images of the same individual from the Gallery images.

We utilized CMC and mAP to evaluate our experimental results. Rank 1 and Rank 5 are often used for two standards of CMC. Take Rank 1 as an example for the explanation of CMC. Given an animal image, we compare the similarity of the Query image and the images in the Gallery image sets and rank the Gallery images according to their similarity. A function is defined to judge whether the two images (q, i) have the same label:(10)$$F\left(l_{q},l_{i}\right)≜\{\begin{array}{c} 1\;\;if\;l_{q}=l_{i}\\ 0\;\;if\;l_{q}\neq l_{i} \end{array}$$

Then, when calculating Rank 1, it is only necessary to count whether all Query pictures are the same as their first returned results.
(11)$$rank\;1=\frac{1}{\left\|Q\right\|}\underset{q\in Q}{\sum }F\left(l_{q},l_{1}^{q}\right)$$

Q is the set of all query pictures, Query, and $l_{1}^{q}$ is the label of the i-th returned result in the image library corresponding to the query picture q.

### 3.2. Effectiveness of the Cross-Attention Module

We replaced the original VIT attention module with the CAB to form a new transformer model. The CAB pays more attention to the differences in animal fur patterns and facial features. Figure 6 shows the attention maps generated by the CAB. As seen in Figure 6, the patterns on the backs of tigers, the prominent features of panda and tiger faces, and the lips and ears of chimpanzees were gradually enhanced through three CAB attention modules.

Since our model can re-identify the body appearances of animals, it is not limited to re-identification based on animal faces. The model proposed by Yu et al. [16] is based on the re-identification of the body appearance of Amur tigers. In addition, the research used three evaluation metrics: Rank 1, Rank 5, and mAP, whereas most of the research models only use mAP. Therefore, we compared our model with Yu et al. and the LA transformer [24] on the Amur tiger dataset in order to more comprehensively evaluate the effectiveness of the cross-attention module. As shown in Figure 7, our method is comparable to Yu et al. for Rank 5. The metrics Rank 1 and Rank 5 of our method are more than 97% and over 99%, respectively, although the metrics Rank 1 and Rank 5 of our method are a bit lower than Yu et al. Thus our method still achieved a satisfying result. Furthermore, the metric mAP of our method is 2.02% higher than Yu et al. Rank 1 is an indicator for the re-identification evaluation, which can represent the quality of the model to some extent. However, Rank 1 only needs to hit the first image, in which some accidental factors exist. Relatively speaking, mAP can more comprehensively reflect the authenticity of the whole model because the mAP value is obtained by calculating average the AP value for all categories. It is worth mentioning that compared with the LA transformer network, our improved model has obvious advantages in three metrics.

### 3.3. Decision on the Layer Number of a Local Aware Network

The layer number of the local aware network directly affects the feature extraction performance of the model. More layers mean fewer differences between features, which will reduce the classification accuracy of the model. As shown in Figure 8, over the course of iterations, the mAP value of the 28-layer structure was shown to be obviously inferior to those of the 7-layer and 14-layer ones, and the mAP value of the 14-layer one was about 91.5%. The 7-layer structure performed best, with a mAP of over 92.5%.

### 3.4. Comparison against the State of the Art

The datasets mainly included four categories of images: body appearance, color face, gray face, and complex face datasets. Since existing studies have conducted experiments on these datasets, in order to fairly evaluate our model from multiple perspectives, we chose the state-of-the-art models corresponding to each type of dataset for the comparison: the Yu et al. model for the ATRW dataset; the Tri-AI model for color face and gray face datasets; and PrimNet [18] for C-Zoo and C-Tai datasets. At the same time, in order to evaluate our models more intuitively, we also chose two re-identification methods CAL + ResNet [30] and Top-DB-Net [31] for the comparison. The two methods focus on extracting and mining the discriminant features in insignificant areas.

As the testing set for the ATRW dataset was not yet released, Yu et al. also used the images in the training set for experiments, which were consistent with ours. Rank 1, Rank 5, and mAP are widely recognized as the standard evaluation metrics in the existing literature. We also used Rank1, Rank5, and mAP for the ATRW dataset, and mAP and Rank 1 for the Tri-AI dataset and the PrimNet dataset, respectively. The metrics and the proportions of the training set and testing set were consistent with these models’ original studies. The experimental results are shown in Table 2.

As shown in Table 2, Top-DB-Net, CAL+ResNet, our method, and Jiwen Yu’s method, achieve very high Rank 1 and Rank 5 in ATRW. Although our method was slightly inferior to CAL + ResNet and Jwen Yu’s method in the first two indicators, our mean average precision was superior to the other two methods. It is worth mentioning that our model has obvious advantages in three indicators compared with Top-DB-Net and LA transformer. In fact, Rank 1 is indeed an important evaluation indicator, but mAP can better reflect the quality of the model, as explained in Section 3.2. For the color face and the gray face datasets, our method obtained the highest evaluation score, followed by Top-DB-Net. In particular, due to nearly the same face features among different individuals in the challenging datasets in the C-Zoo and C-Tai, all the models faced difficulties in recognizing the animal IDs. Despite all this, our method had more than a 10% higher mAP than the other two methods. Although the hit rate of the first picture of CAL + Resnet was the highest, the Rank 1 value of our method also achieved over 85%.

Through the experiments on the above datasets, we proved that our network has good generalization ability and can be well applied to various re-identification tasks with wild terrestrial animals. The reason for this is that our model extracts the local features of small image patches and uses the cross-attention module to calculate the correlation characteristics between patches in order to construct the global information. For some wild animal species, their fur patterns and facial features have distinctive differences, so the prediction results are satisfactory. However, the faces of different chimpanzee individuals are almost the same except for their lips and ears, which increases the difficulty in identifying different individuals.

## 4. Conclusions

With the invasion of human society to nature, the living space of wild animals is gradually decreasing. To help understand the living conditions of wild animals, we designed a novel transformer-based re-identification method for wild terrestrial animals which incorporates the cross-attention mechanism and the locally aware transformer. In addition, we fine-tuned the hierarchical structure of the locally aware network to accommodate wildlife body structures. We evaluated the model on various datasets. The experimental results showed that our model had superior performance in identifying animal individuals than the existing state-of-the-art methods, even for challenging datasets.

In general, the performance of wildlife re-identification using the proposed methods significantly improved. However, not all animals’ coats have stripes or spots, which greatly increases the difficulty of animal re-identification recognition. Similar to other existing methods, our model also suffered from the problem of being unable to extract subtle differences in the faces of species with a solid fur color and similar faces, such as chimpanzees. In addition, due to the occlusion of wild vegetation, the camera only can capture a part of the animal’s body. The images can also be blurred by the influence of noise and bad weather such as strong winds, heavy fog and rain. Therefore, more in-depth research is needed for animal re-identification recognition.

In future research, we will try to use counterfactual attention learning [30] to learn more effective attention based on causal inference, which can measure attention quality and develop a powerful supervisory signal to guide the learning process. We believe that the attention scheme can better learn the subtle features of animal individuals. At the same time, we will consider using the target detection network to identify and segment the animals in order to reduce the influence of backgrounds. And fully consider the individual identification of translocated species [32].

## Figures and Tables

**Figure 1 animals-12-03503-f001:**
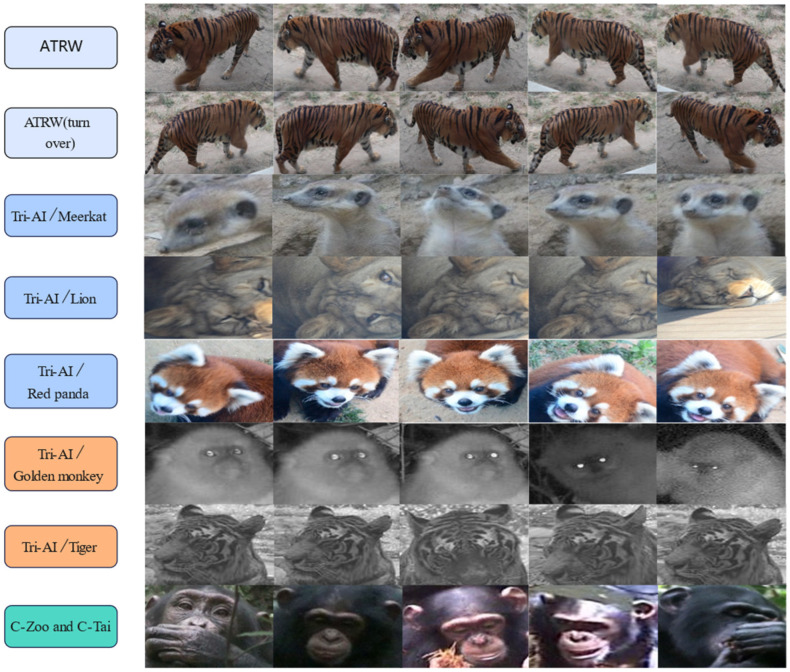
Examples of images from the dataset.

**Figure 2 animals-12-03503-f002:**
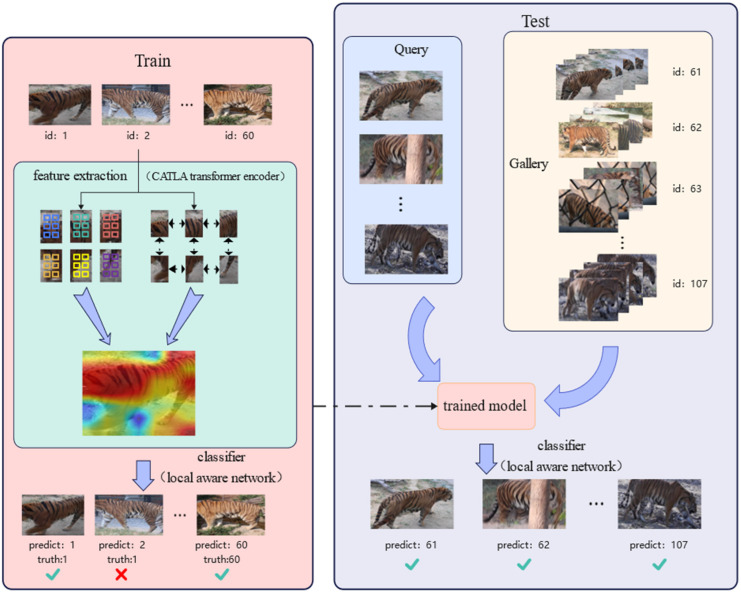
Workflow of the wildlife re-identification.

**Figure 3 animals-12-03503-f003:**
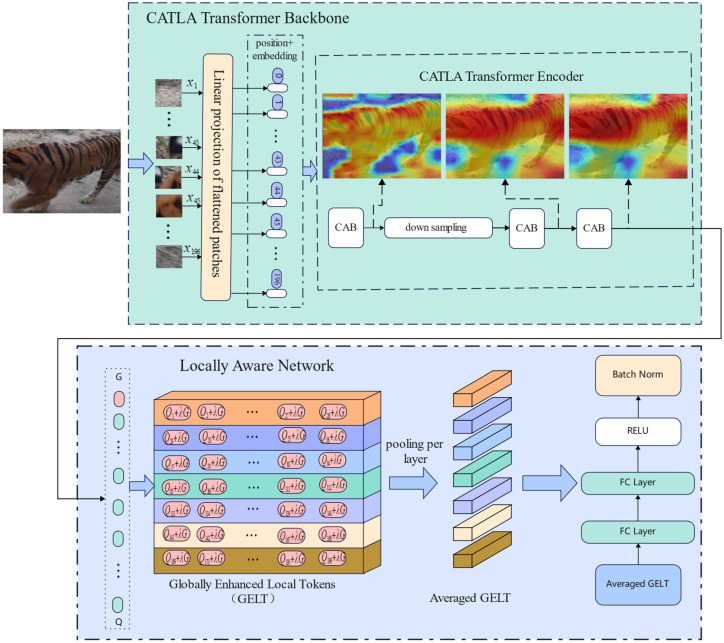
The overall pipeline of our approach (the input image is 224 × 224, and the patch is 16 × 16, so 196 local tokens will be generated). The generated tokens F = f_0, f_1, …, f_196, where f_0 is the global token. The global token is called G, and the local tokens are called Q. The global and local tokens are combined using a weighted-average method and arranged into a 7 × 28 2D distribution. Each layer is pooled and finally fed into the classifier.

**Figure 4 animals-12-03503-f004:**
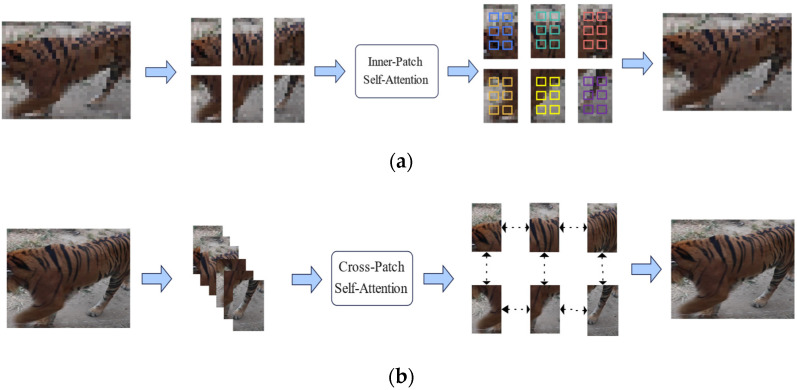
Principles of the IPSA module and CPSA module: (**a**) IPSA module; and (**b**) CPSA module.

**Figure 5 animals-12-03503-f005:**
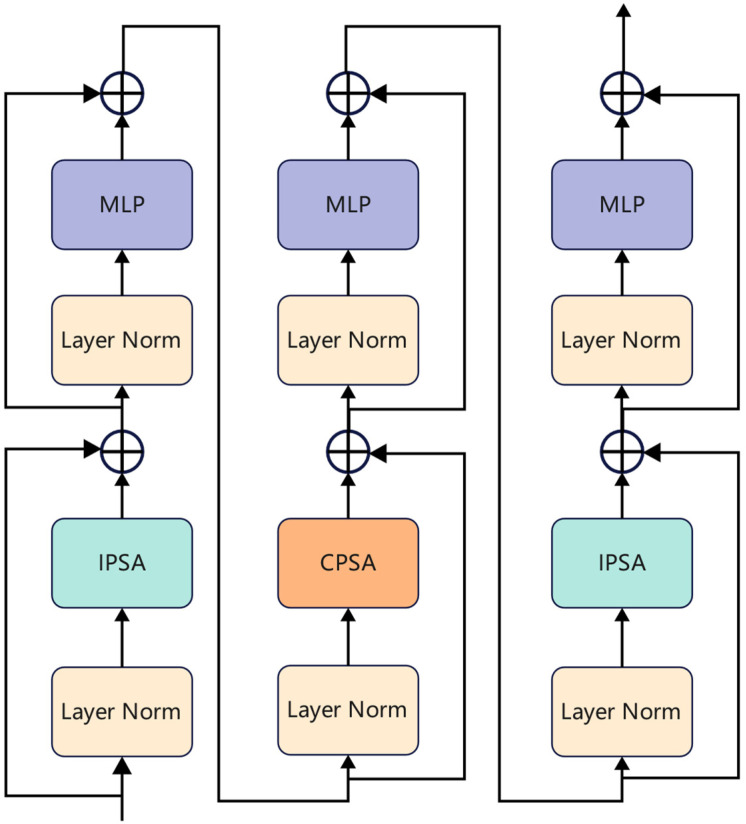
Cross-attention module stacking mode.

**Figure 6 animals-12-03503-f006:**
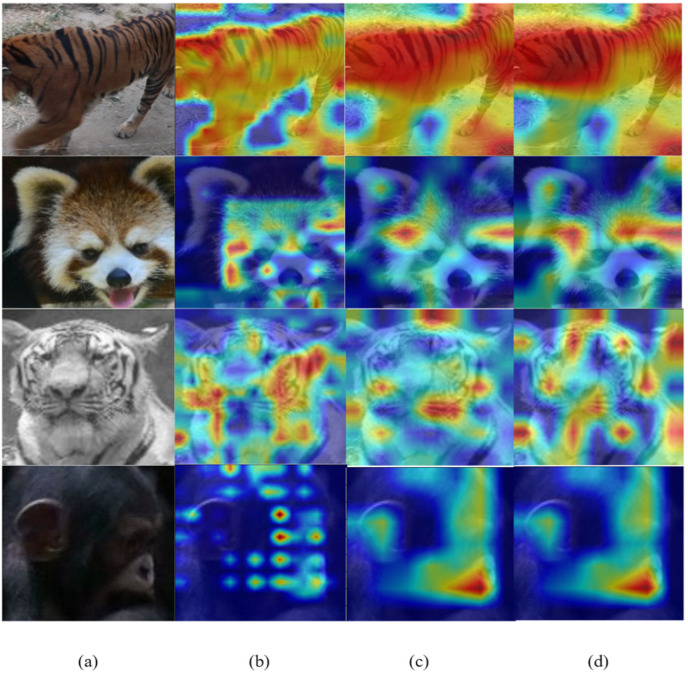
Animal attention hotspot map: (**a**) the original image; (**b**) the attention map after the first CAB block; (**c**) the attention map after the second CAB block; and (**d**) the attention map after the third CAB block.

**Figure 7 animals-12-03503-f007:**
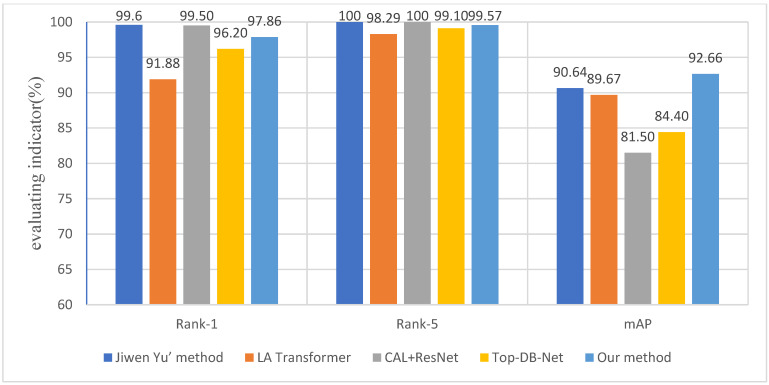
Experimental comparison results between the model with cross-attention and other models.

**Figure 8 animals-12-03503-f008:**
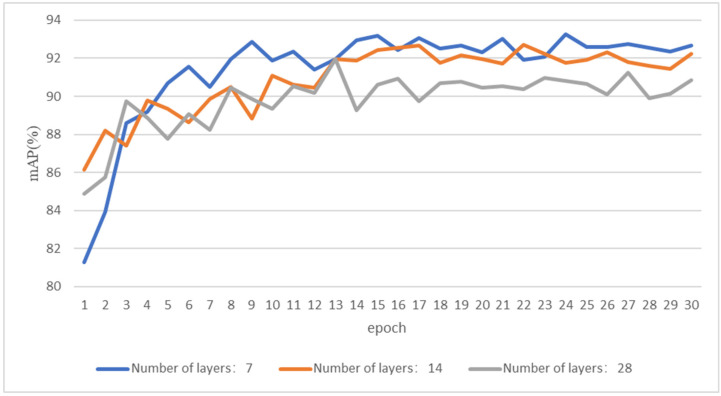
mAP values of the different layer structures.

**Table 1 animals-12-03503-t001:** Statistics of the training set and test set for the re-ID task.

Datasets		Training Set		Testing Set			
		Train Ids	Train Images	Query Ids	Query Images	Gallery Ids	Gallery Images
Tiger body	ATRW	120	2310	47	234	47	499
Tri-AI Color face	Meerkat	12	480	6	14	6	155
	Lion	11	402	6	48	6	231
	Red panda	11	369	5	51	5	231
Tri-AI Gray face	Golden monkey	17	287	7	12	7	57
	Tiger	13	603	6	56	6	147
Complex face	C-Zoo and C-Tai	116	4917	50	419	50	2346

**Table 2 animals-12-03503-t002:** Experimental results of different methods.

Datasets	Species	Methods	Rank 1	Rank 5	mAP
ATRW	Tiger	Jiwen Yu’s method	**99.6**	**100**	90.64
		CAL + ResNet	**99.5**	**100**	81.50
		Top-DB-Net	96.20	99.10	84.40
		LA Transformer	91.88	98.29	89.67
		Ours	97.86	99.57	**92.66**
Tri-AI Color face	Meerkat	Tri-AI	-	-	90.13
		CAL + ResNet	-	-	93.90
		Top-DB-Net	-	-	95.40
		Ours	-	-	**96.16**
	Lion	Tri-AI	-	-	93.55
		CAL + ResNet	-	-	90.20
		Top-DB-Net	-	-	95.50
		Our method	-	-	**98.69**
	Red Panda	Tri-AI	-	-	92.16
		CAL + ResNet	-	-	90.30
		Top-DB-Net	-	-	95.20
		Ours	-	-	**98.51**
Tri-AI Gray face	Golden monkey	Tri-AI	-	-	94.38
		CAL + ResNet	-	-	92.40
		Top-DB-Net	-	-	96.40
		Our method	-	-	**97.99**
	Tiger	Tri-AI	-	-	92.03
		CAL + ResNet	-	-	78.90
		Top-DB-Net	-	-	92.40
		Ours	-	-	**93.77**
Complex face	chimpanzee	PrimNet	75.82	-	-
		CAL + ResNet	**89.70**		63.70
		Top-DB-Net	83.30		55.60
		Ours	85.03	-	**73.79**

## Data Availability

The data presented in this study are available on request from the corresponding author.

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
