# Peer review of "Wild Terrestrial Animal Re-Identification Based on an Improved Locally Aware Transformer with a Cross-Attention Mechanism"

_animals, 2022, doi:10.3390/ani12243503_

Round 1
Reviewer 1 Report
When I looked at Animals Journals Content (https://www.mdpi.com/authors/layout#_bookmark57), I saw that the article did not include Material Methods and Results. I suggest you review the author guidelines again.
Line 48: IUCN ?
Line 57-59: Camera traps form the scientific basis of this work. More information about camera traps (Which models should be more suitable? How should the camera traps be placed in the stations in order to get more effective results? It is useful to add detailed information.
INTRODUCTION
Individual identification of wild animals with the effective methods specified in the application is also very necessary for determining population size and densities. For this reason, it would be appropriate to include the literature on the usability of the individual diagnosis method used in this study in population size and density determination.
Why only endangered species? This study is also important as it will contribute to the individual identification of spotted and striped species. Information on this topic can be added.
Lines 97-112: It is appropriate to write this section under the heading "Materials and Methods". That's why the "Datasets" heading should be placed under "Materials and Methods".
Supporting the formulas and models used in the study with the literature will increase the quality of the study.
The "Conclusion" section is only 13 lines. I suggest that this section be developed and expanded and the study findings discussed further.
Reviewer 2 Report
The paper proposes an interesting methodology, however I think that a journal paper must present in a more formal way the proposed method. In that sense the paper needs some refinements.
Please make a deeper state-of-the-art, as in references [8] and [22] which include a more extensive comparison, and give a comparison with respect to other two methods of the state-of-the-art.
Please in Table 1, indicate what part of the images correspond to the testing data set, here explain what percentage was used as Trainig, and as testing? In literature it is recomended to use 70% for training and 30% for testing.
Your visual explanation of the proposed method from figure 2 to figure 5, is very ilustrative, however it is important to present the method in a more formal way, describing the main equations of the whole procedure.
Please explain a little bit more, why to calculate ranks 1 and 5, if only compares with the ATRW and Complex face databases? In figure 6 I conclude that the best method is the Yu´s method, two of three metrics are better.
Please make a deep revision of the english, for example, is better to write: Some standard evaluation metrics in literature are used such as... in the palce of: We used the same evaluation metrics as the original papers...
Round 2
Reviewer 2 Report
This second version of the paper has been significantly improved.
All my concerns have been addressed.